# Single-Switch Inverter Modular Parallel Multi-Voltage Levels Wireless Charging System for Robots

**DOI:** 10.3390/s26010067

**Published:** 2025-12-22

**Authors:** Hua Li, Zhiyuan Sun, Lianfu Wei

**Affiliations:** 1School of Information Science and Technology, Southwest Jiaotong University, Chengdu 611756, China; lihua@my.swjtu.edu.cn; 2Senad Technology Co., Ltd., Shanghai 201815, China; 3School of Electrical Engineering, Southwest Jiaotong University, Chengdu 611756, China; sunzhiyuan@my.swjtu.edu.cn

**Keywords:** wireless charging (WC), single-switch inverter, multi-voltage levels, modular parallel, constant current and voltage outputs

## Abstract

With the continuous development of the robotics industry, using a single wireless system to charge different types of robots has become a critical issue that urgently needs to be addressed. To solve this problem, in the present work, we propose a single-switch inverter module wireless charging system based on parallel module number frequency modulation to achieve the expected variable voltage output by adjusting the operating frequency and the number of parallel modules, thereby enhancing the interoperability between devices. To meet the charging requirements of lithium batteries, which require constant current (CC) first and constant voltage (CV) thereafter, we first discuss how to implement CC and CV charging modes, then demonstrate that the proposed system can provide the required CC and CV output under various load conditions. Subsequently, a simplified equivalent circuit model to achieve this wireless charging system is proposed and an exact expression for its equivalent input voltage source is provided. Subsequently, based on the analysis of the amplitude–frequency characteristics of voltage gain under the CV mode, we propose the relevant method and program to realize this variable output system, and specifically build a prototype system based on a three-module parallel configuration. Experimental results show that the present prototype system can indeed provide the constant current (CC) and constant voltage (CV) outputs required for lithium battery charging, and the expected variable voltage output achieved by frequency modulation (FM) is verified. Its maximum efficiency can approach 91.3%. Compared with other wireless charging systems with single-switch inverters, this prototype experimental system possesses significant advantages in completing the full charging process of lithium batteries, maintaining stable voltage output during the constant voltage phase, and enabling flexible multi-voltage output.

## 1. Introduction

Compared to wired charging, wireless charging (WC) possesses certain advantages, such as high safety, high flexibility, and ease of operation, as the usual physical connections through wires are unnecessary [1,2,3]. Therefore, WC has already been widely applied in various fields such as kitchen appliances, automated underwater vehicles (AUVs) [4], inspection drones [5], automated guided vehicles (AGVs) [6,7], electric vehicles (EVs) [8,9,10,11,12], and robots [13,14,15], etc.

Generically, depending on the type of high-frequency inverters used, WC can be roughly classified into the three modes, including the full-bridge (FB) inverter, half-bridge (HB) inverter, and single-switch resonant inverter wireless charging (SSRIWC). Typically, the FB and HB inverters have been widely applied in various scenarios due to their relatively low voltage stress on the switches and low harmonic distortion in the inverter waveform [16,17,18,19]. However, their circuit structures present certain practical drawbacks, mainly including the risk of arm shoot-through, a high number of switches, and complex control [20,21]. The circuit in a single-switch resonant inverter consists of only one switching device, thus resulting in fewer components with this reduced volume. Nevertheless, its extension of power capacity is limited, since its power transfer ability relies on resonance and the switching device is subjected to high voltage stress. To overcome such limitations, this study aims to integrate the high power handling capability of full-bridge and half-bridge inverters with the advantages of a single-switch topology, such as a reduced number of devices and lower breakdown risk, for designing and experimentally implementing a lithium-ion battery wireless charging system, with a superior overall performance, to meet the demands of multiple application scenarios.

Although robots have been widely adopted in daily life with the development of artificial intelligence (AI) technology, the batteries of different types of robots require varying charging voltages due to diverse working conditions and the requirements of different charging system parameters and structures. If various robot models could utilize a unified wireless charging system, it would significantly enhance charging convenience. For example, with inductive power transfer (IPT) coils, reference [22] demonstrated matched charging for different devices, which can eliminate the potential hazards caused by incorrect charging. Other studies focused on multi-level current and voltage outputs [21,22,23,24,25,26] by incorporating additional passive or active components and specialized control methods in circuits. Reference [27] achieved the desired output through parameter optimization of bilateral LCC networks, while reference [28] implemented multi-voltage output by adding two Buck converters on the secondary side. However, these approaches increase the sizes of systems and may compromise safety. Although reference [29] attained a wide-range voltage output and zero voltage switch (ZVS) by adjusting the number of parallel modules, operation modes, and frequency, their system still required additional transformers and capacitors and the control of multiple switching devices also presented certain challenges [30]. Nevertheless, the proposed method still achieved a significant reduction in the number of MOSFETs compared with the approaches reported in other studies. The concept of frequency modulation (FM) and parallel unit control further expands the voltage regulation range, prevents excessive voltage rise during charging, and improves the safety of systems. Even under identical control, driving, and communication conditions, FB or HB inverters inherently require additional dead time [31] and complex synchronous-drive coordination. These mechanisms inevitably introduce extra timing delays and uncertainties. In contrast, the single-switch multi-parallel inverter does not require dead-time insertion, synchronous drive control, or complementary gate-signal management. Therefore, under the same control-signal conditions, the proposed single-switch structure minimizes the influence of communication-induced timing delays. Basically, the absence of shoot-through risks may enhance safety, and the aforementioned literature can provide valuable inspiration.

In order to preserve the advantages of the SSRIWC system in the process of modular parallel operation, enabling robots at different voltage levels to achieve WC with the same charging device, thereby significantly improving convenience, this paper proposes a modular parallel single-switch inverter power system capable of wirelessly charging various types of robots.

The main contributions of this work are listed as follows:
(1)By combining sensor-based scanning with software-controlled FM, this paper addresses the common limitation of existing WC systems that can only charge specific devices and cannot accommodate batteries of various device types. The proposed method requires no modification to the circuit structure, which significantly reduces the size and cost of the charging equipment and greatly improves the interoperability of the charging system.(2)This paper proposes an SSRIWC system which—compared with the widely used FB or HB inverter circuits—retains the advantages of single-switch circuits, such as fewer MOSFETs and a smaller size, while avoiding a substantial increase in the MOSFET voltage stress, thus offering certain benefits.(3)The system achieves automatic switching from CC to CV mode during lithium-battery charging through battery-voltage sampling and relay control. Moreover, a fine frequency-tuning mechanism is employed to address the issue of voltage overshoot during the CV stage that typically occurs in open-loop wireless charging systems.


The remaining contents of this paper are organized as follows: Section 2 conducts modeling and analysis of the proposed system structure, including CC output characteristics, CV output characteristics, and derivation of the system’s equivalent voltage source. Section 3 analyzes the frequency-dependent characteristics of system voltage gain, proposes an FM method for the system, and provides the corresponding process flowchart. Section 4 establishes a three-parallel experimental prototype and conducts relevant experiments. The prototype output obtained in the experiments agrees well with the theoretical analysis, confirming that variable voltage output can be realized by FM. Section 5 gives a conclusion to the whole paper.

## 2. The Principle of the Proposed Wireless Charging System

The proposed SSMPC system is shown in Figure 1. This article uses the simulation software saber launched by Synopsys to simulate the operating state of the system. *U*_dci_ is the DC input voltage of the *i*-th parallel module (*i* represents the *i*-th parallel module, 1 ≤ *i* ≤ n, where n is the number of parallel modules in the system), *C*_ini_ is the DC input filter capacitor, *Q*_i_ is the power MOSFET, and *D_Q_*_i_ is the body diode of the MOSFET, and *L*_ri_ and *C*_ri_ are the *LC* parallel resonant inductor and capacitor, respectively, used to maintain the regular operation of the SSC. Inter-Cell Transformer (ICT_i_) is the current-sharing (CS) inductor, while *L*_pri_ and *C*_pri_ are the current-increasing (CI) inductor and capacitor. *L*_p_ is the primary-side coil inductance, *C*_p_ is the primary-side compensation capacitor, and *r*_p_ is the primary-side coil internal resistance. *L*_s_ is the secondary-side coil inductance, and *C*_s_ is the tuning capacitor, which is in series with *L*_s_ to adjust the inductance of the secondary-side coil and the circuit gain. *C*_s1_ and *C*_s2_ are the CC/CV switching compensation capacitors, *L*_s1_ is the secondary-side compensation inductor, and *M* represents the mutual inductance of the loosely coupled transformer (LCT). *r*_s_ is the internal resistance of the secondary-side coil. *R*_load_ is the equivalent battery resistance. The actual battery voltage *G*_bat_ is sampled (with the voltage gain *G* used to represent the corresponding voltage in the following steps), and the battery’s cutoff charging voltage *G*_e_ is input as an analog signal into the MCU ADC module. The CC to CV switching point gain is selected as *G*_s_ = 0.95*G*_e_. When *G*_s_ is reached, the topology switches to CV mode. At the same time, *G*_e_ and *G*_bat_ are sent to the primary-side digital comparator. The comparator provides a comparison signal to drive the PWM generator to produce an FM signal. When the battery voltage rises close to *G*_e_, the charging process is complete. During CC charging, the switching frequency remains unchanged. FM is only used in CV mode to achieve multiple voltage-level outputs. 

### 2.1. Analysis of Constant Current Mode in the Circuit

Given that lithium batteries must undergo a CC stage followed by a CV stage during charging, in this section, the output voltage and current expressions of the system in both CC and CV modes are derived.

Further analysis can be conducted by simplifying the circuit using the ICT equivalent model from [32].

The fundamental-frequency equivalent model of the circuit topology shown in Figure 1 is illustrated in Figure 2, where *R*_eq_ denotes the equivalent resistance of the load resistor *R*_load_ after the rectifier bridge [33], satisfying the following:(1)$$R_{\mathrm{load}}=\frac{\pi^{2}}{8}R_{\mathrm{eq}}$$

After neglecting the internal resistances of other components and incorporating capacitor *C*_s1_ into the circuit, the equivalent model of the circuit in CC mode can be represented by the T-network model shown in Figure 3, where *L*_sum_ and *C*_pr_ satisfy the following relationship:(2)$$\omega L_{\mathrm{sum}}=\frac{1}{\omega C_{\mathrm{pr}}}$$

Clearly, they are both in complete resonance. 

Under the condition specified by Equation (3), Figure 3a can be simplified through circuit transformation to the minimal form shown in Figure 3d.(3)$$\left\{\begin{array}{l} j\omega C_{p}+\frac{1}{j\omega (L_{p}-M)}=0\\ j\omega L_{s}^{\prime }=j\omega L_{s1}=j\omega \left(L_{s}-\frac{1}{\omega^{2}L_{s}}\right)\\ j\omega L_{s}^{\prime }+\frac{1}{j\omega C_{s1}}=0 \end{array}\right.$$

The output current is as follows:(4)$${\dot{I}}_{o}=\frac{n{\dot{U}}_{\mathrm{in}}M}{j\omega L_{\mathrm{sum}}L_{s}}$$

Formula (4) indicates that the output current depends only on the fundamental equivalent voltage ${\dot{U}}_{\mathrm{in}}$, the number of parallel units n, the mutual inductance *M*, the equivalent total inductance *L*_sum_, the angular frequency *ω*, and the receiving-end inductance *L*_s_, and is independent of the equivalent load resistance *R*_eq_. It is demonstrated that the system can achieve load-independent CC output.

### 2.2. Analysis of Constant Voltage Mode in the Circuit

When the capacitor *C*_s2_ is connected to the circuit, the circuit can be simplified to the model shown in Figure 4.

*C*_s_´ represents the parallel equivalent capacitance of *L*_s1_ and *C*_s2_. To achieve CV output in the circuit, Equation (5) should be satisfied, as follows:(5)$$\left\{\begin{array}{l} \frac{j\omega L_{s1}\cdot \frac{1}{j\omega C_{s2}}}{j\omega L_{s1}+\frac{1}{j\omega C_{s2}}}=\frac{1}{j\omega C_{s}^{\prime }}\\ \omega L_{s}^{\prime }=\frac{1}{\omega C_{s}^{\prime }} \end{array}\right.$$

Based on this, the equivalent parallel capacitors *C*_s_´ and *C*_s2_ satisfy Equation (6), as follows:(6)$$\left\{\begin{array}{l} C_{s}^{\prime }=\frac{\omega^{2}L_{s1}C_{s2}-1}{\omega^{2}L_{s1}}\\ C_{s2}=\frac{L_{s1}+L_{s}}{\omega^{2}L_{s1}L_{s}^{\prime }}=\frac{2}{\omega^{2}L_{s}^{\prime }} \end{array}\right.$$

The output voltage is as follows:(7)$${\dot{U}}_{o}=\frac{n{\dot{U}}_{\mathrm{in}}M}{L_{\mathrm{sum}}}$$

Formula (7) indicates that the output voltage depends only on the number of parallel units *n*, the fundamental equivalent voltage ${\dot{U}}_{\mathrm{in}}$, mutual inductance *M*, and the equivalent total inductance *L*_sum_, and is independent of the equivalent load resistance *R*_eq_. It is demonstrated that the system can achieve load-independent CV output.

### 2.3. Analysis of Equivalent Voltage Source

Since the CI–LC-type secondary-side compensation network proposed in this paper is highly complex in its equivalent form, the simplified CI–LC–SS topology of the charging system is analyzed to facilitate calculation. The circuit is shown in Figure 5a, where *C*_eq_ = *nC*_r_. The analysis of other compensation networks can be extended based on the above procedure, indicating that the proposed method has a certain degree of generality.

Reference [31] analyzed the single-switch circuit using a complex frequency-domain model but did not consider the impact of varying the number of parallel modules (*n*) on system performance. Building upon previous research, this paper investigates the influence of parallel module count on the output, thereby addressing the limited modeling accuracy in earlier studies. Moreover, it provides a general analytical method for multi-parallel single-switch inverter CI–LC wireless charging systems, which can be readily extended to systems employing different secondary compensation networks.

Through the circuit transformations and equivalences shown in Figure 5, the minimal equivalent circuit illustrated in Figure 5f is finally obtained. The transformation process satisfies Equation (8).(8)$$\left\{\begin{array}{l} Z_{\mathrm{eq}}=\frac{{\left(\omega M\right)}^{2}}{j\omega L_{s}+\frac{1}{j\omega C_{s}}+R_{\mathrm{eql}}}\\ Z_{\mathrm{eq}1}=\frac{1}{j\omega C_{p}}+j\omega L_{p}+Z_{\mathrm{eq}}\\ Z_{\mathrm{eq}2}=\frac{1}{j\omega C_{\mathrm{sum}}}//Z_{\mathrm{eq}1}\\ Z_{\mathrm{eq}3}=Z_{\mathrm{eq}2}+j\omega \frac{L_{\mathrm{sum}}}{n}\\ Z_{\mathrm{eq}4}=Z_{\mathrm{eq}3}//j\omega \frac{L_{r}}{n} \end{array}\right.$$

To perform complex frequency-domain equivalence on the CI–LC–SS type circuit depicted in Figure 5c, the resulting model is shown in Figure 6.

The MOSFET is designated as State 1 when conducting and State 2 when off. Analyzing the models for State 1 and State 2, we derive the time-domain models as shown in Figure 6a,c, respectively, which satisfy the following:(9)$$\left\{\begin{array}{l} U_{\mathrm{dc}}=L_{eq}\frac{di_{L_{eq1}}(t)}{dt}+i_{L_{eq1}}(t)R\\ u_{C_{eq}}(t)=L_{eq}C_{eq}\frac{d^{2}u_{C_{eq}}(t)}{dt^{2}}+RC_{eq}\frac{du_{C_{eq}}(t)}{dt} \end{array}\right.$$

Their corresponding complex frequency-domain models are illustrated in Figure 6b,d. Writing circuit equations for Figure 6b leads to Equation (10), as follows:(10)$$\frac{U_{\mathrm{dc}}}{s}+L_{\mathrm{eq}}I_{1}=I_{L_{\mathrm{eq}1}}(s)\left(sL_{\mathrm{eq}}+R\right)$$

Analysis of the circuit shown in Figure 6d yields Equations (11) and (12), as follows:(11)$$\frac{U_{\mathrm{dc}}}{s}+L_{\mathrm{eq}}I_{2}=I_{L_{\mathrm{eq}2}}(s)\left(\frac{1}{sC_{\mathrm{eq}}}+sL_{\mathrm{eq}}+R\right)$$(12)$$U_{C_{\mathrm{eq}}}(s)=I_{L_{\mathrm{eq}2}}(s)(sL_{\mathrm{eq}}+R)-L_{\mathrm{eq}}I_{2}$$

Assuming the initial current value satisfies Equation (13), as follows:(13)$$\left\{\begin{array}{c} i_{L_{\mathrm{eq}}1}(0_{-})=I_{1}\\ i_{L_{\mathrm{eq}}2}(0_{-})=I_{2} \end{array}\right.$$

Combining Equations (10)–(13), we obtain expressions for $I_{L_{\mathrm{eq}1}}(s)$ and $I_{L_{\mathrm{eq}2}}(s)$, and after applying the inverse Laplace transform, their time-domain expressions are given as shown in Formula (14), as follows:(14)$$\left\{\begin{array}{l} i_{L_{\mathrm{eq}1}}(t)=\frac{U_{\mathrm{dc}}}{R}-\frac{(I_{2}R-U_{\mathrm{dc}})e^{-2\tau t}}{R}\\ i_{L_{\mathrm{eq}2}}(t)=e^{-\tau t}[I_{1}\cosh(\delta t)+\frac{2U_{\mathrm{dc}}-I_{1}R}{2L_{\mathrm{eq}}\delta }\sinh(\delta t)] \end{array}\right.$$
where(15)$$\left\{\begin{array}{l} \tau =\frac{R}{2L_{\mathrm{eq}}}\\ \delta =\frac{\sqrt{C_{\mathrm{eq}}(R^{2}C_{\mathrm{eq}}-4L_{\mathrm{eq}})}}{2C_{\mathrm{eq}}L_{\mathrm{eq}}} \end{array}\right.$$

For the simplest equivalent model of the single-switch circuit shown in Figure 6c, the circuit operational waveform is depicted in Figure 7.

During the period 0–*t*_1_, the MOSFET is turned on, and the inductor current varies approximately linearly.

During *t*_1_–*t*_5_, the MOSFET is turned off, and *u_C_*_eq_(*t*) and *i_L_*_eq_(*t*) exhibit an underdamped resonant behavior, varying approximately in a sinusoidal form.

During *t*_5_–*t*_6_, the system operates in the ZVS state, where *u_C_*_eq_(*t*) has resonated to *U*_dc_, while *i_L_*_eq_(*t*) remains approximately in a linear variation.

Where the *t*_ZVS_ is as follows:(16)$$t_{\mathrm{ZVS}}=(1-D)T-t_{r}$$

The following relationship should be satisfied by analyzing the current $i_{L_{\mathrm{eq}}}$ flowing through the equivalent inductance, as depicted in Figure 6.(17)$$\left\{\begin{array}{l} i_{L_{\mathrm{eq}1}}(0)-i_{L_{\mathrm{eq}2}}(T-DT)=0\\ i_{L_{\mathrm{eq}1}}(DT)-i_{L_{\mathrm{eq}2}}(0)=0 \end{array}\right.$$

From Equations (13) and (14), *I*_1_ and *I*_2_ can be solved as follows:(18)$$\left\{\begin{array}{l} I_{1}=\frac{e^{\tau \left(D-1\right)T}U_{dc}\left\{\left[2L_{eq}\delta \cosh(\delta (D-1)T)(e^{-2\tau DT}-1)]+R\sinh(\delta (D-1)T)(e^{-2\tau DT}+1\right)\right\}}{R\left\{2L_{eq}\delta \left[e^{-\tau (D+1)T}\cosh\left(\delta \left(D-1\right)T\right)-1\right]+e^{-\tau \left(D+1\right)T}\sinh\left(\delta \left(D-1\right)T\right)\right\}}\\ I_{2}=\frac{2U_{dc}\left\{L_{eq}\delta \left[e^{-2\tau DT}-1\right]+e^{-\tau \left(D+1\right)T}\sinh\left[\delta \left(D-1\right)T\right]R\right\}}{R\left\{2L_{eq}\delta \left[e^{-\tau (D+1)T}\cosh\left(\delta \left(D-1\right)T\right)-1\right]+e^{-\tau \left(D+1\right)T}\sinh\left(\delta \left(D-1\right)T\right)\right\}} \end{array}\right.$$

The expression for the current $i_{L_{\mathrm{eq}}}(t)$ flowing through the equivalent inductor over the entire time domain can be derived from Equations (14) and (18), as follows:(19)$$i_{L_{\mathrm{eq}}}(t)=\left\{\begin{array}{l} i_{L_{\mathrm{eq}2}}(t),(k-1)T<t<(1-D-D_{\mathrm{ZVS}})kT\\ i_{L_{\mathrm{eq}1}}(t),(1-D-D_{\mathrm{ZVS}})kT<t<kT \end{array}\right.$$

The expression for $U_{C_{\mathrm{eq}}}(s)$ in the complex frequency domain can be obtained based on Equation (12). By performing the inverse Laplace transform, its time-domain expression is derived as follows, where *k* = 1,2,3…(20)$$u_{C_{\mathrm{eq}}}(t)=\left\{\begin{array}{c} e^{-\tau t}(U_{\mathrm{dc}}\cosh(\delta t)+\frac{C_{\mathrm{eq}}RU_{\mathrm{dc}}-2L_{\mathrm{eq}}I_{2}}{2\delta C_{\mathrm{eq}}L_{\mathrm{eq}}}\sinh(\delta t),(k-1)T<t<(1-D-D_{\mathrm{ZVS}})kT\\ U_{\mathrm{dc}},(1-D-D_{ZVS})kT<t<kT \end{array}\right.$$

To obtain the AC fundamental component, a Fourier decomposition is performed, resulting in Equation (21), as follows:(21)$$u_{C_{\mathrm{eq}}}(t)=A_{0}+A_{1}\cos(\omega t+\varphi )+\sum_{i=2}^{\infty }A_{i}\cos(i\omega t+\varphi )$$

*A*_0_ represents the DC component of the waveform, *A*_1_ denotes the amplitude of the fundamental frequency, and *A_i_* represents the amplitude of the *i*-th harmonic. The DC and fundamental components satisfy Equation (22), as follows:(22)$$\left\{\begin{array}{l} A_{0}=a_{0}\\ A_{1}=\sqrt{a_{1}^{2}+b_{1}^{2}} \end{array}\right.$$

The coefficients *a*_0_, *a*_1_ and *b*_1_ satisfy the following:(23)$$\left\{\begin{array}{l} a_{0}=\frac{1}{T}\int_{0}^{T}u_{C_{\mathrm{eq}}}(t)dt\\ a_{1}=\frac{2}{T}\int_{0}^{T}u_{C_{\mathrm{eq}}}(t)\cos(\omega t)dt\\ b_{1}=\frac{2}{T}\int_{0}^{T}u_{C_{\mathrm{eq}}}(t)\sin(\omega t)dt \end{array}\right.$$

## 3. Analysis of Voltage Gain and Frequency Modulation

During the operation of a single-switch circuit, the relationship between system voltage gain and frequency changes very closely. This section will elaborate and analyze it.

### 3.1. Voltage Gain Analysis

The CV mode T-network shown in Figure 4 can be simplified to the model in Figure 8. The impedances in this network are defined as follows in Equation (24):(24)$$\left\{\begin{array}{l} Z_{1}=j\omega (L_{s}^{\prime }-M)+\frac{1}{j\omega C_{s}^{\prime }}\\ Z_{2}=j\omega M//(Z_{1}+R_{\mathrm{eq}})\\ Z_{3}=j\omega (L_{p}-M)+\frac{1}{j\omega C_{p}}\\ Z_{4}=\frac{1}{j\omega C_{\mathrm{sum}}}//(Z_{2}+Z_{3}) \end{array}\right.$$

The voltage phasor should satisfy the following relationship:(25)$$\left\{\begin{array}{l} {\dot{U}}_{2}=\frac{Z_{2}}{Z_{2}+Z_{3}}{\dot{U}}_{1}\\ {\dot{U}}_{1}=\frac{Z_{4}}{j\omega \frac{L_{\mathrm{sum}}}{n}+Z_{4}}{\dot{U}}_{\mathrm{in}}\\ {\dot{U}}_{3}=\frac{R_{\mathrm{eq}}}{Z_{1}+R_{\mathrm{eq}}}{\dot{U}}_{2}\\ U_{o}=\frac{\pi }{2\sqrt{2}}U_{3} \end{array}\right.$$

The voltage gain *G*_v_ satisfies the following:(26)$$G_{v}=\frac{U_{o}}{U_{\mathrm{dc}}}$$

From Equations (25) and (26), the expression for voltage gain *G*_v_ is obtained as follows:(27)$$G_{v}=\left|\frac{\sqrt{2}n\pi R_{\mathrm{eq}}Z_{2}Z_{4}}{4(Z_{1}+R_{\mathrm{eq}})(Z_{2}+Z_{3})(j\omega L_{\mathrm{sum}}+nZ_{4})}\right|$$

From Equation (27), the voltage gain curve shown in Figure 9 can be derived. The output voltage varies significantly under different numbers of parallel units and frequencies. This provides a basis for charging robots with varying voltage and power levels by adjusting the number of *n* and *f*.

Figure 10 demonstrates the gain under changing load conditions. It can be observed that when the system operates at the resonant frequency of 85 kHz, the gain is the same under different loads, verifying that the system can maintain a constant voltage characteristic at 85 kHz.

### 3.2. System Frequency Modulation Scheme

Figure 11 and Figure 12, respectively, show the flowchart and schematic diagram of system frequency modulation. First, regarding the selection of the operating frequency range, since there is currently no globally unified standard for robotic wireless charging, and considering that robots and electric vehicles share similar wireless charging mechanisms and basic circuit structures, we refer to the SAE J2954 wireless charging standard for electric vehicles. Accordingly, the operating frequency range of 80–90 kHz is selected for the system. Within this range, different frequency levels *f*_1_… *f_x_*… *f*_s_ are chosen (where 1 ≤ *x* ≤ s, *f_x_* is the *x*-th variable level point, and *f*_s_ is the maximum frequency level point). The selection of s requires balancing software computational resources and voltage regulation accuracy, and an appropriately moderate value is sufficient. Each level point is chosen as a specific range near it, denoted as (*f*_1min_, *f*_1max_)… (*f*_smin_, *f*_smax_). The load ID is scanned to obtain *G*_e_, which is then compared with the sampled battery voltage *G*_bat_ to determine whether the battery is fully charged. If not fully charged, the system output voltage *G*_v_ should be compared with the expected charging voltage *G*_e_ for frequency modulation. If *G*_v_ < *G*_e_, the frequency needs to be determined. If the frequency is within the *x*-th level interval, it can be increased; if it is not present, we determine whether the *x*-th interval is the last. If not, we switch to the next interval to continue increasing the frequency. If so, this means that all frequencies under the number of parallel connections cannot meet the requirements, and the number of parallel connections needs to be increased to achieve a larger output voltage. When *G*_v_ > *G*_e_, the analysis process is the opposite. When *G*_v_ = *G*_e_, this indicates that the system output voltage is suitable for charging the battery. Under normal conditions, the system should continuously maintain *G*_v_ = *G*_e_ until *G*_bat_ gradually rises with the charging process and reaches the desired charging voltage, at which point the charging is complete. The modulation process, Figure 12, is consistent with Figure 11, which shows the changes in frequency and number of parallel connections for achieving *G*_e_ output voltage under the initial conditions of *G*_v1_ and *G*_v2_.

Equation (28) presents the formula for calculating the system efficiency.

Here, *P*_o_ represents the output power, and the different subscripts denote the losses associated with different components. The efficiency analysis is based on the calculation of active power.(28)$$\eta =\frac{P_{o}}{P_{\mathrm{LCT}}+\sum_{i=1}^{n}\left({L_{\mathrm{sumi}}}^{2}r_{L\mathrm{sumi}}+{L_{L\mathrm{ri}}}^{2}r_{L\mathrm{ri}}\right)+P_{Ls1}+P_{o}}$$

The power losses of the high-frequency inverter comprise the conduction loss *P*_cond_MOS_ of the MOSFETs, the conduction loss *P*_cond_bd_ of the body diodes, and the switching loss *P*_SW_ of the MOSFETs and the diodes. Nevertheless, the proposed *P*_cond_MOS_ and *P*_cond_bd_ are given by the following [34]:(29)$$\left\{\begin{array}{l} P_{\mathrm{cond}\_\mathrm{MOS}}=\frac{1}{4\pi }r_{\mathrm{DS}}I_{\mathrm{in}}^{2}(\pi +\varphi +\sin\varphi )\\ P_{\mathrm{cond}\_\mathrm{bd}}=\frac{\sqrt{2}}{2\pi }V_{f}I_{\mathrm{in}}[1-\sin(\frac{\varphi }{2})]+\frac{1}{\pi }r_{D}I_{\mathrm{in}}^{2}(\pi -\varphi -\sin\varphi ) \end{array}\right.$$
where *r*_DS_ is the drain–source on-state resistance of the MOSFET, *I*_in_ is the current flowing through the MOSFET, *φ* is the phase-shift angle of the inverter, *V*_f_ is the threshold voltage, and *r*_D_ is the equivalent conduction resistance of the body diode.

*P*_SW_ is expressed by the following [35]:(30)$$P_{\mathrm{SW}}=\frac{\sqrt{2}}{2}U_{\mathrm{dc}}\left|I_{\mathrm{in}}f\sin(\theta )\right|(t_{r}/3+t_{f}/2)$$
where this is the phase-shift angle of the inverter, *f* is the switching frequency, θ is the phase angle between voltage and current, and $t_{r}$ and $t_{f}$ are the rise time and fall time of the MOSFET, respectively.

Therefore, the power loss of the inverter $P_{\mathrm{inv}\_\mathrm{loss}}$ is given by the following:(31)$$P_{\mathrm{inv}\_\mathrm{loss}}=P_{\mathrm{cond}\_\mathrm{MOS}}+P_{\mathrm{cond}\_\mathrm{bd}}+P_{\mathrm{SW}}$$

Let $P_{\mathrm{inv}\_\mathrm{out}}$ denote the active power at the inverter output. Then, the inverter efficiency is given by the following:(32)$$\left\{\begin{array}{l} P_{\mathrm{inv}\_\mathrm{out}}=U_{\mathrm{DS}}I_{\mathrm{in}}\cos(\theta )\\ \eta =\frac{P_{\mathrm{inv}\_\mathrm{out}}}{P_{\mathrm{inv}\_\mathrm{out}}+P_{\mathrm{inv}\_\mathrm{loss}}} \end{array}\right.$$

## 4. Experimental Verification

A single-switch three-parallel prototype was designed to verify the analysis and modulation method proposed in this paper, with output voltage levels of 220 V and 330 V. The experimental setup is shown in Figure 13. The operating temperature of the prototype is 25 °C.

As shown in Figure 13, the main parts of the prototype consist of a DC input, single-switch inverters, CI–LCs, ICTs, a compensation capacitor, a magnetic coupler, a receiving-side circuit, an electronic load, and receiving and transmitting Bluetooth, as well as the control circuits for both the primary and secondary sides and a battery scanner on the receiving side. The switching transistors are SiC MOSFETs (NVHL040N120SC1), driven by PWM signals generated by STM32F030K6T6. The rectifier diodes are C3D20060. The electronic load IT8616 is used for lithium batteries, and an LCR digital bridge (model ZX8532) is used to measure the parameters of the magnetic components at the rated operating frequency of 85 kHz. The coils are wound with 0.1 mm × 700-strand Litz wire, with a current rating of 27.48 A. The measurements show that the circuit model parameters are in high agreement with those of the actual prototype.

The basic parameters of the system in normal operating conditions are shown in Table 1.

System parameters are shown in Table 2.

The parameters of other circuit components are listed in Table 3.

Table 4 presents the electrical and environmental characteristics of the system’s resonant components. From this, it can be seen that for CBB capacitors and alloy-core inductors, the resonance point shift caused by component tolerances and temperature drift is minimal and within the allowable error range, which validates the reasonableness of the previously derived formulas.

Figure 14 shows the waveforms of the MOSFET’s *V*_DS_ and *V*_GS_ when *n* = 3, *f* = 85 kHz, and *R* = 50 Ω. The results indicate that the system MOSFET operates under ZVS conditions with a sufficient margin, demonstrating the appropriateness of the parameter design.

In a real charging process, the equivalent load of a battery typically varies continuously, and the charging of a lithium battery requires a constant-current (CC) stage followed by a constant-voltage (CV) stage. Figure 15 illustrates the variation in the equivalent battery load simulated using an electronic load. It can be observed that even when the equivalent battery load undergoes a sudden change—which represents the most severe load variation—the system is still able to maintain constant-current and constant-voltage outputs. This demonstrates that the proposed system exhibits strong load-independent CC and CV output characteristics.

Figure 16 presents the closed-loop time-domain response of the system at a 330 V output. The system performs continuous sampling at 50–100 kHz and employs a second-order Butterworth low-pass filter for signal conditioning. The ADC resolution is 12 bits, and the total delay of the digital control loop is approximately 5–100 μs. The corresponding time-domain parameters are shown in the Figure 16, where *T*_s_ denotes the settling time and σ% represents the percentage overshoot. As illustrated, when the load undergoes variations of 50 Ω → 100 Ω → 50 Ω, the closed-loop controller is able to regulate the output voltage precisely to 330 V, demonstrating an excellent voltage regulation performance.

As shown in Figure 17, during the CC stage, the output current remains nearly constant at approximately 10.2 A, demonstrating that the system provides a load-independent constant-current output at this operating frequency. When the equivalent load increases to around 32 Ω, the system transitions from the CC mode to the CV mode. Although the output voltage shows a slight increase during this stage, it can still be regarded as approximately constant within an acceptable range, thereby confirming that the system exhibits a load-independent constant-voltage output at this frequency.

Figure 18 and Figure 19 show the relationship between the switching frequency, output voltage, and current of the system at the 220 V and 330 V levels. It can be seen that the battery equivalent load increases progressively during charging. The system can vary the output voltage over a wide range via frequency modulation, enabling charging of different lithium batteries. Moreover, slight adjustments of the switching frequency allow for stable voltage output, demonstrating the effectiveness of the proposed frequency-modulation approach for variable voltage regulation.

Figure 20 presents the efficiency curves during the charging process at the following two voltage levels: (a) represents the 220 V output, and (b) represents the 330 V output. It can be observed that as the charging process progresses, the efficiency initially increases and then decreases with the equivalent resistance of the battery. The maximum efficiency is achieved at the CC to CV transition points (22 Ω and 32 Ω).

Figure 21 presents the temperature distribution of the components after operating at the rated power for a period of time. It can be observed that the maximum device temperature is approximately 42 °C, which is not excessively high and does not pose a risk to system safety.

Figure 22 presents the loss breakdown of the proposed system under constant-voltage outputs of 220 V and 330 V. The analysis primarily includes the losses of the transmitter and receiver coils, MOSFETs, magnetic components, and rectifier bridge. It can be observed that, under all operating conditions, the losses of the magnetic coupler coils account for the largest portion, mainly because the coil currents are relatively high. The second major contributors include the MOSFETs, magnetic components, and the rectifier bridge.

All loss measurements were obtained by measuring the RMS current flowing through each component and calculating the corresponding loss using its ESR or parasitic resistance. The loss measurements were obtained by using an Agilent N2783B current probe to measure the RMS current, which was then multiplied by the corresponding ESR or parasitic resistance to calculate the associated losses.

Table 5 presents a performance comparison between the proposed prototype and those reported in other references. It can be seen that the proposed prototype can achieve the complete constant-current and constant-voltage charging process required for lithium batteries, realize variable voltage output through frequency modulation, and reach a maximum efficiency of 91.3%. Table 6 presents a comparison of the output and safety performance between the proposed system and those reported in previous studies. It can be seen that the proposed system demonstrates clear advantages over existing works in terms of voltage regulation range, mitigation of voltage overshoot, and system safety.

By considering Table 5 and Table 6 together, it can be seen that the proposed system shows improvements in all aspects listed in the tables compared with existing studies, and the overall performance of the proposed prototype demonstrates clear superiority.

By comparing the variable output in Table 5 and the voltage regulation range in Table 6, the extra components and MOS/inverter in Table 5, the CC/CV indicator in Table 5, and the voltage overshoot and system safety in Table 6, we demonstrate the validity of the three contributions stated in the introduction, thereby strengthening the technical soundness of this work.

## 5. Conclusions

This paper proposes a single-switch inverter modular parallel system based on frequency and parallel-module number modulation, which achieves variable voltage output by adjusting the operating frequency and the number of parallel modules to accommodate the charging requirements of different types of lithium batteries, providing a solution for one-to-many wireless charging. Through analysis of the system under different operating modes, it is demonstrated that a complete charging process for lithium batteries can be realized. At the same time, based on the existing literature, a more general modeling method is proposed, which effectively reflects the output performance and equivalent model changes under different parallel module counts. Similar system modeling can be extended based on this method. Furthermore, based on the analysis of the voltage-gain variation curve in the CV mode, a modulation scheme involving frequency and parallel-module number is proposed, and the implementation flowchart is provided. Finally, a three-module parallel prototype is constructed for validation. Experimental results show that the system can achieve the CC and CV outputs required for lithium battery charging, and that variable voltage output can be realized via FM, confirming the correctness of the preceding analysis. Comparison with other related studies further demonstrates the advantages of the proposed system. In addition, the proposed system still has the problem of a large number of passive components on the receiving side. In the future, reducing the volume of the receiving side by using control or hybrid compensation topologies will be considered.

## Figures and Tables

**Figure 1 sensors-26-00067-f001:**
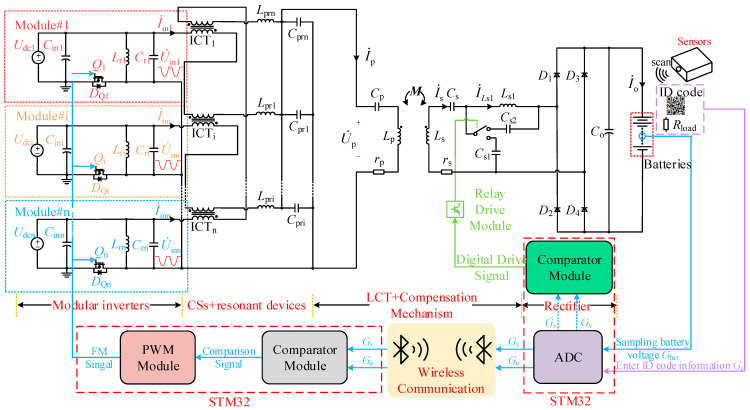
The proposed SSMPC system.

**Figure 2 sensors-26-00067-f002:**
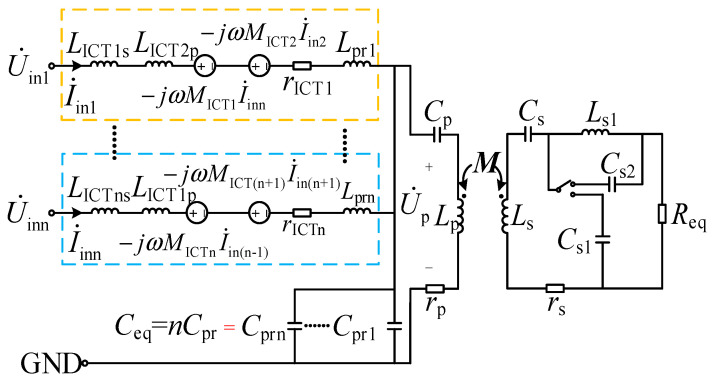
The fundamental harmonic equivalent model of the proposed topology.

**Figure 3 sensors-26-00067-f003:**
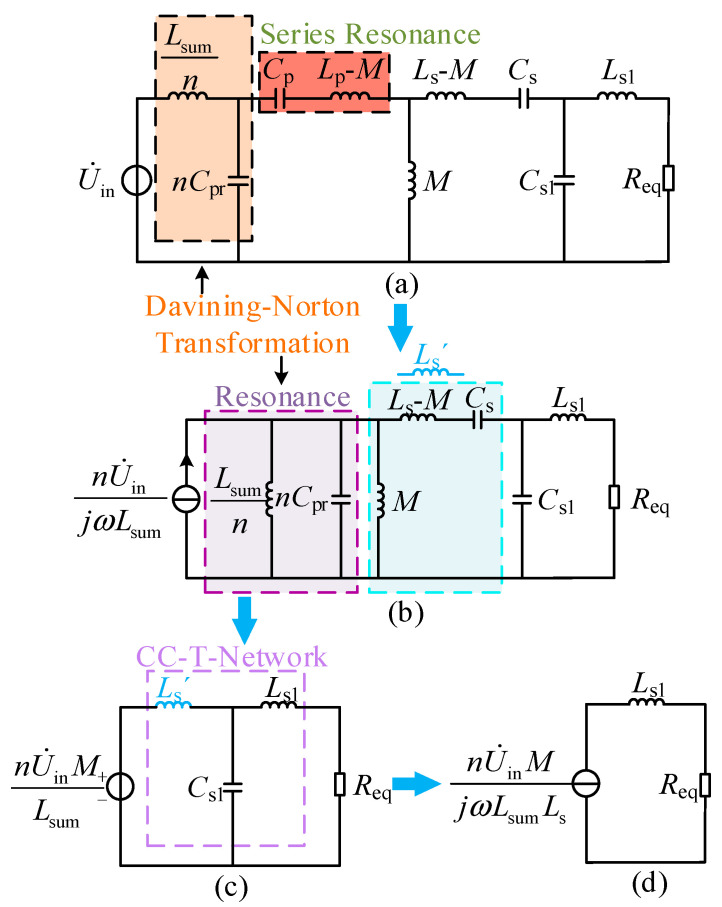
The equivalent T-network in constant current mode. (**a**) CC model; (**b**) initial Davining-Norton equivalent; (**c**) CC T-network equivalent; (**d**) minimal CC model.

**Figure 4 sensors-26-00067-f004:**
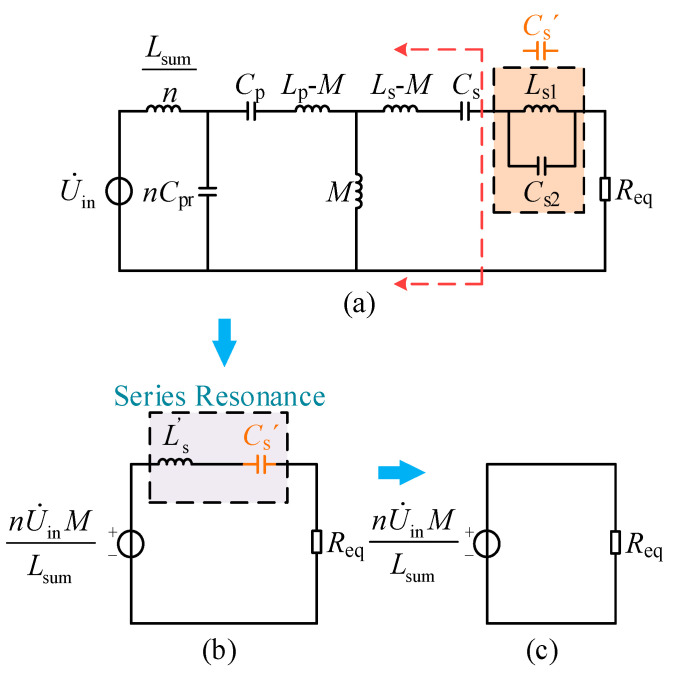
The equivalent T-network in constant voltage mode. (**a**) CV model; (**b**) secondary-side equivalent circuit; (**c**) minimal CV model.

**Figure 5 sensors-26-00067-f005:**
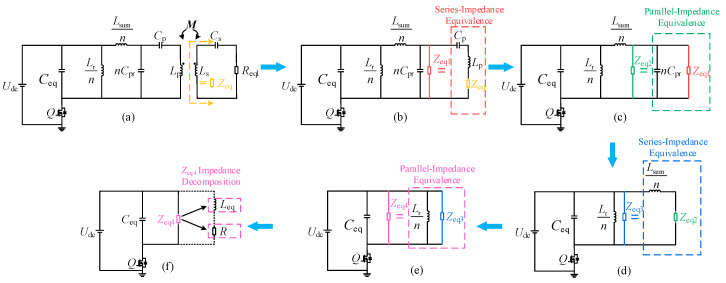
CI–LC–SS-type WC circuit and its equivalent model. (**a**) initial CI–LC–SS model; (**b**) secondary-side equivalent model; (**c**) series equivalent model; (**d**) parallel equivalent model; (**e**) series equivalent model; (**f**) impedance decomposition model.

**Figure 6 sensors-26-00067-f006:**
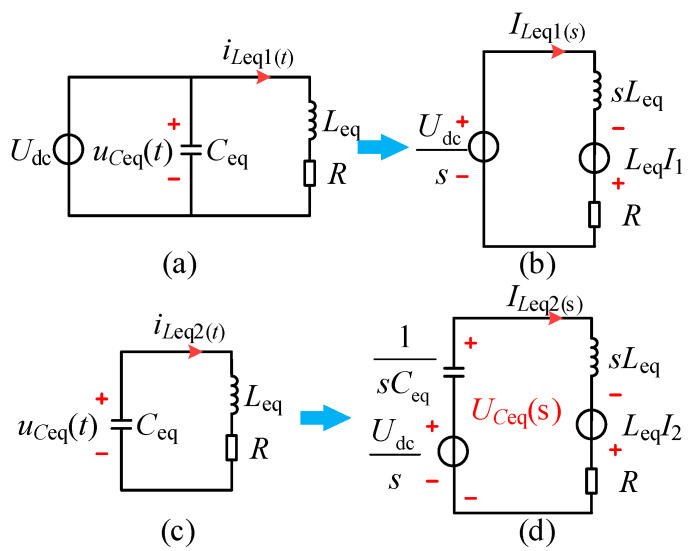
Complex frequency-domain equivalent models. (**a**) Time-domain model for State 1. (**b**) Complex frequency-domain model for State 1. (**c**) Time-domain model for State 2. (**d**) Complex frequency-domain model for State 2.

**Figure 7 sensors-26-00067-f007:**
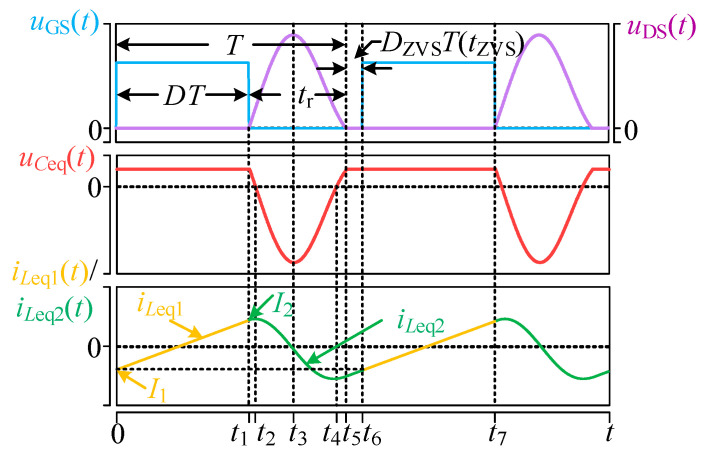
Single-switch circuit operation waveform.

**Figure 8 sensors-26-00067-f008:**
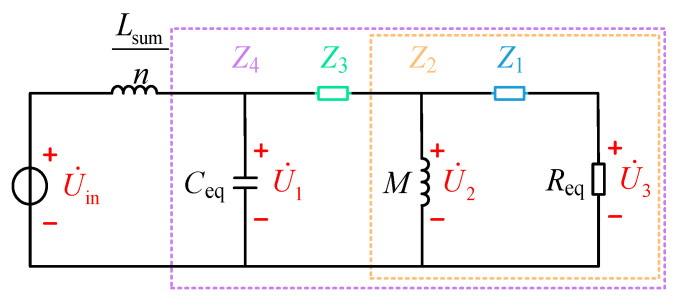
Simplified model of the constant voltage mode T-network.

**Figure 9 sensors-26-00067-f009:**
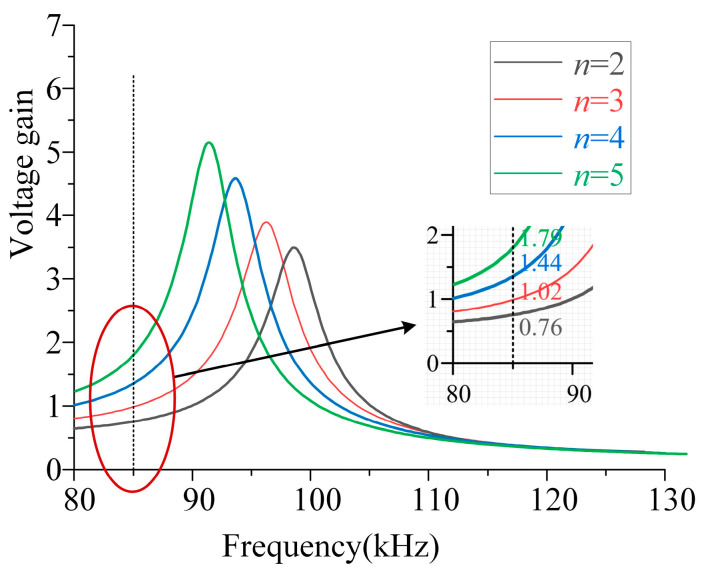
CV mode voltage gain–frequency curve.

**Figure 10 sensors-26-00067-f010:**
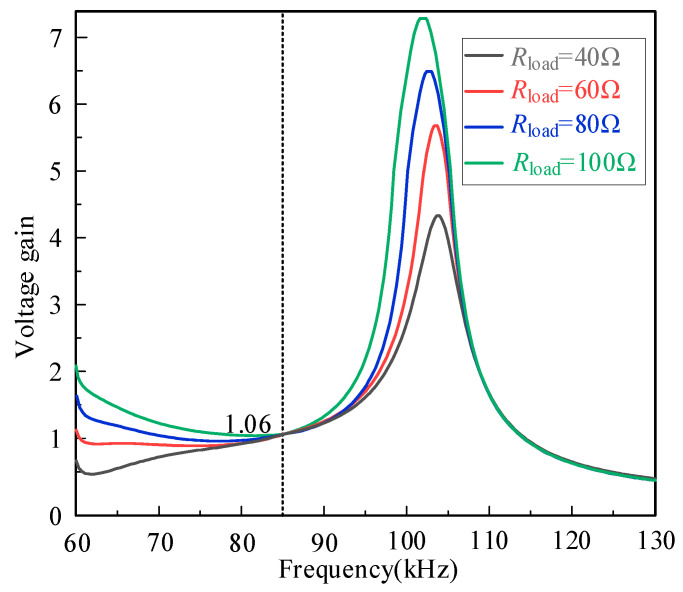
The voltage gain curves of different load when the number of modules is equal to 3.

**Figure 11 sensors-26-00067-f011:**
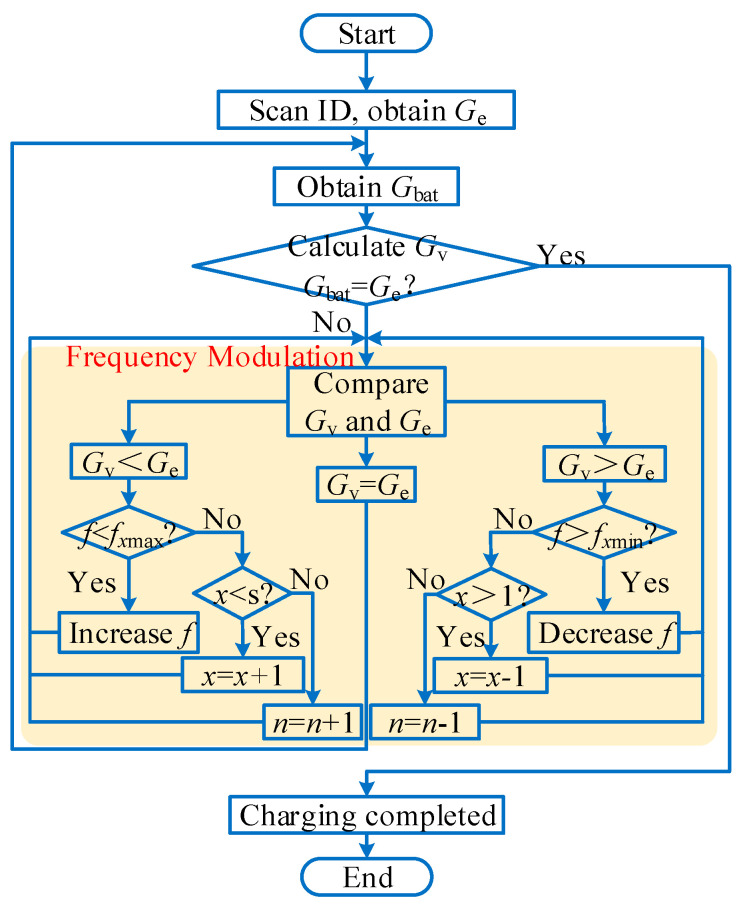
System frequency modulation flowchart.

**Figure 12 sensors-26-00067-f012:**
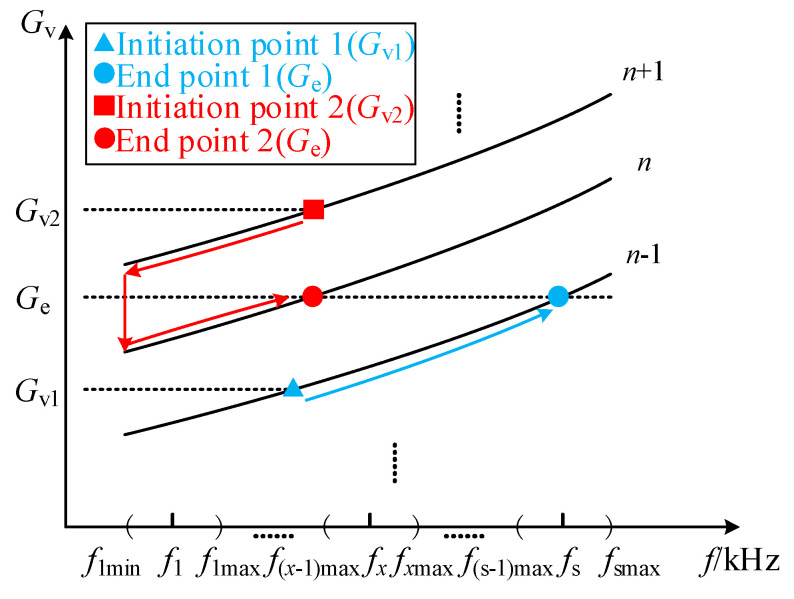
Schematic diagram of frequency modulation.

**Figure 13 sensors-26-00067-f013:**
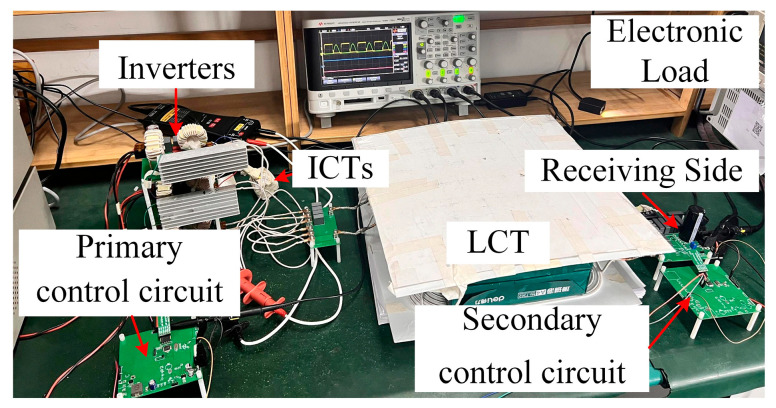
Proposed three-parallel prototype.

**Figure 14 sensors-26-00067-f014:**
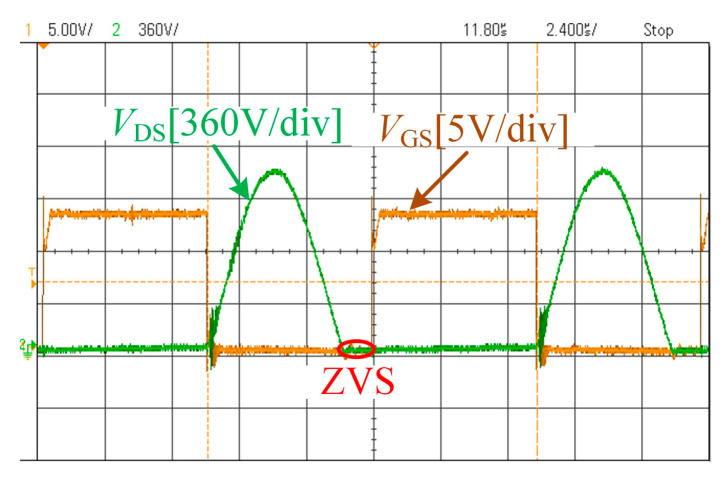
The waveforms of *V*_GS_ and *V*_DS_.

**Figure 15 sensors-26-00067-f015:**
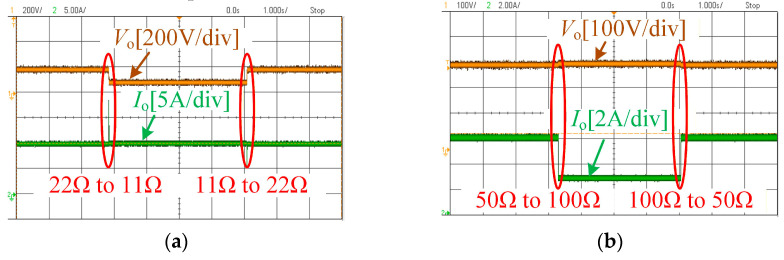
System open-loop dynamic characteristics: (**a**) CC Mode and (**b**) CV Mode.

**Figure 16 sensors-26-00067-f016:**
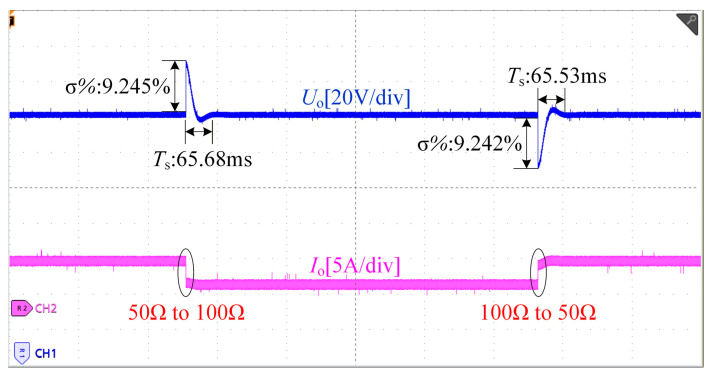
Closed-loop time-domain response of the system.

**Figure 17 sensors-26-00067-f017:**
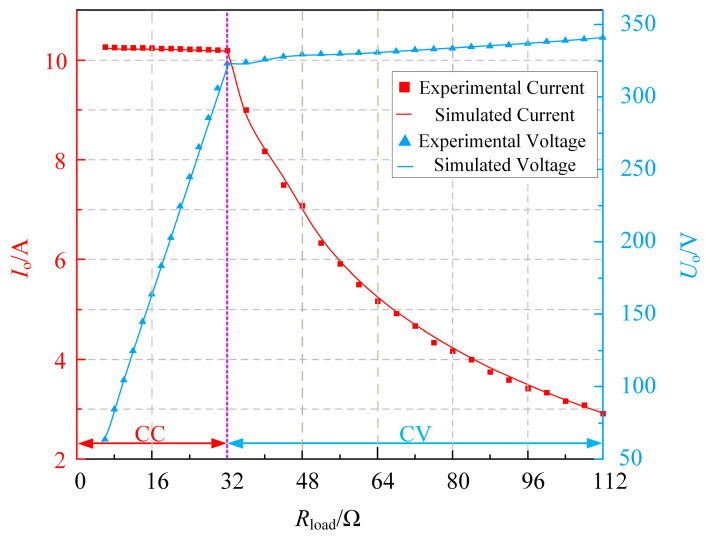
Equivalent battery charging experiment.

**Figure 18 sensors-26-00067-f018:**
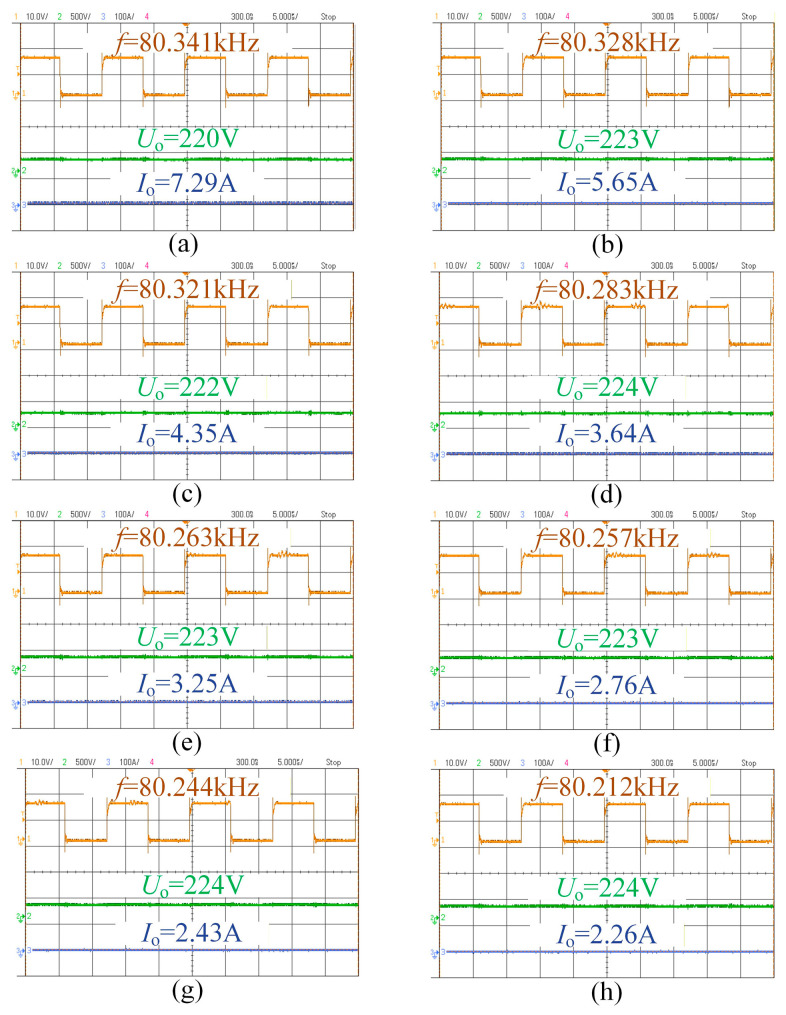
The complete process diagram of CV charging at 220 V: (**a**) *R* = 30 Ω; (**b**) *R* = 40 Ω; (**c**) *R* = 50 Ω; (**d**) *R* = 60 Ω; (**e**) *R* = 70 Ω; (**f**) *R* = 80 Ω; (**g**) *R* = 90 Ω; and (**h**) *R* = 100 Ω.

**Figure 19 sensors-26-00067-f019:**
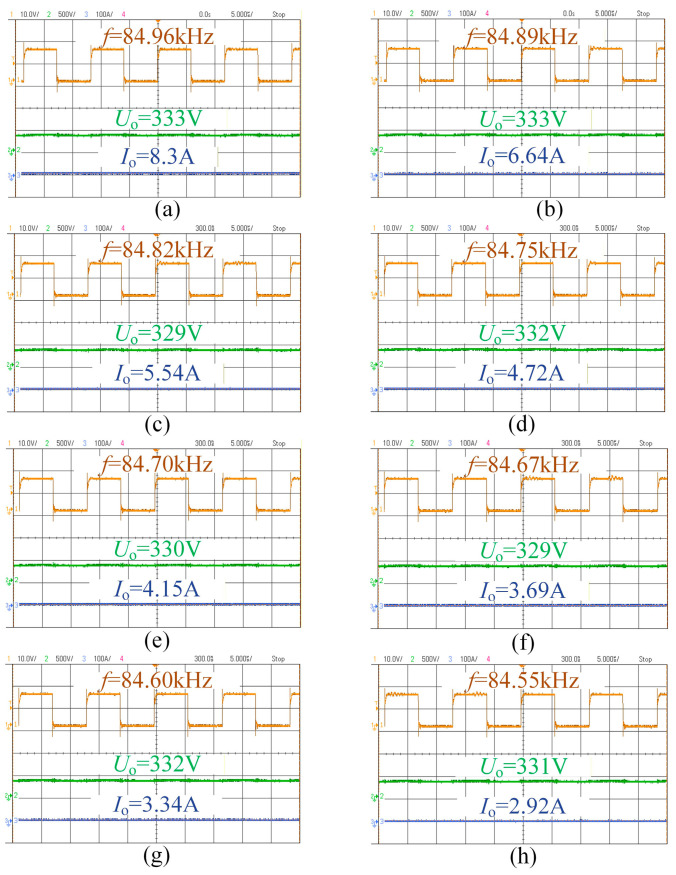
The complete process diagram of CV charging at 330 V: (**a**) R = 40 Ω; (**b**) R = 50 Ω; (**c**) R = 60 Ω; (**d**) R = 70 Ω; (**e**) R = 80 Ω; (**f**) R = 90 Ω; (**g**) R = 100 Ω; (**h**) R = 112 Ω.

**Figure 20 sensors-26-00067-f020:**
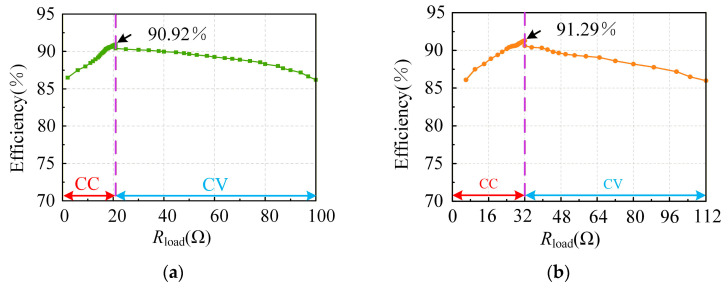
System charging efficiency curves: (**a**) 220 V output and (**b**) 330 V output.

**Figure 21 sensors-26-00067-f021:**
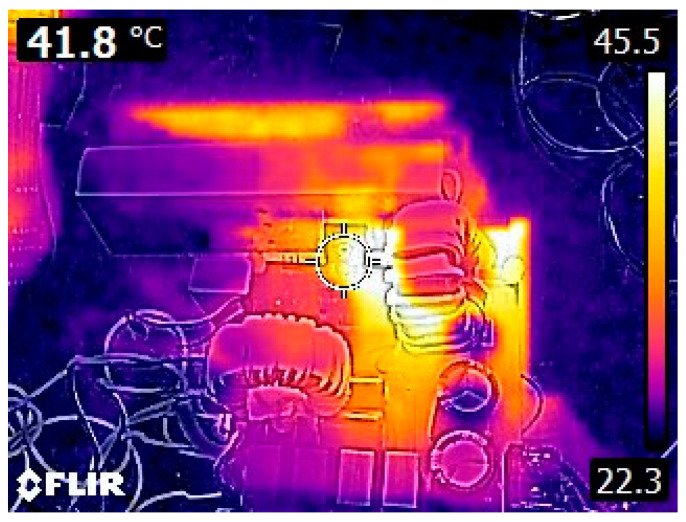
System device temperature.

**Figure 22 sensors-26-00067-f022:**
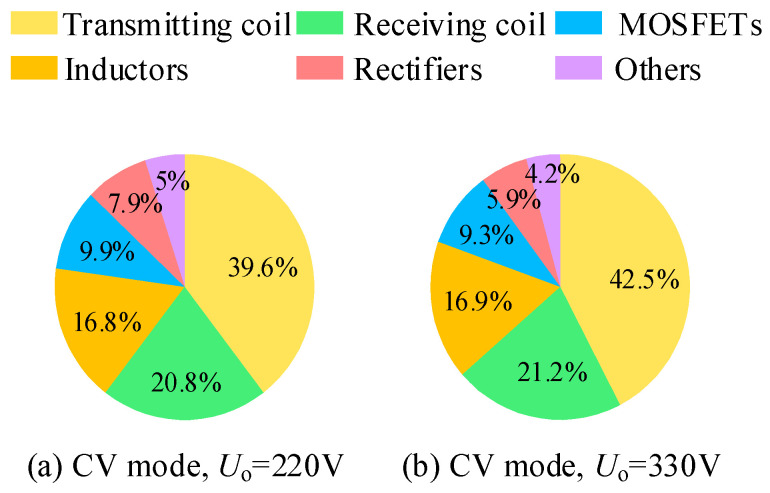
System loss analysis.

**Table 1 sensors-26-00067-t001:** System of operating parameters.

Symbol	Definition	Value
*U*_dc_/*V*	Input DC voltage	311
*f*/kHz	Operating frequency	80.4/85
*V*_o_/*V*	Output voltage	220/330
*D*	Duty cycle	50%
*P*_o_/kW	Output power	2.2/3.3

**Table 2 sensors-26-00067-t002:** Parameters of magnetic couplers and ICT coils.

Symbol	Definition	Value
*h*	Transmission distance	150 mm
*L* _p_	Inductance of the transmitter coil	94.12 μH
*L* _s_	Inductance of the receiving coil	94.05 μH
*M*	Mutual inductance between Lp and Ls	22.06 μH
*r* _p_	Parasitic resistance of Lp	150.98 mΩ

**Table 3 sensors-26-00067-t003:** Parameters of circuits.

Symbol	Value	Symbol	Value	Value	Value
*L* _r1_	60.01 μH	*r_L_* _r3_	59.89 mΩ	*C* _s_	71.99 nF
*L* _r2_	60.05 μH	*L* _sum1_	75.96 μH	*L* _s1_	44.95 μH
*L* _r3_	59.97 μH	*L* _sum2_	75.99 μH	*C* _s1_	81.02 nF
*C* _r1_	30.11 nF	*L* _sum3_	76.13 μH	*C* _s2_	162.11 nF
*C* _r2_	30.06 nF	*C* _pr1_	44.06 nF	*r_L_* _sum1_	54.06 mΩ

**Table 4 sensors-26-00067-t004:** Electrical and environmental characteristics of resonant components.

Symbol	Type	Tolerance	Operating Temperature	Thermal Drift	Rated Parameter
*L*_r1_-*L*_r3_	Alloy toroidal inductor	±8%	−40 °C~+125 °C	2–3%	25 A
*C*_r1_-*C*_r3_	CBB	±5%	−40 °C~+105 °C	Within ±5%	2 kV
*L*_pr1_-*L*_pr3_	Alloy toroidal inductor	±8%	−40 °C~+125 °C	2–5%	5 A
*C*_pr1_-*C*_pr3_	CBB	±5%	−40 °C~+105 °C	Within ±5%	2 kV
*C* _s_	CBB	±5%	−40 °C~+105 °C	Within ±5%	2 kV
*L* _s1_	Alloy toroidal inductor	±8%	−40 °C~+125 °C	About 5%	14 A
*C* _s1_	CBB	±5%	−40 °C~+100 °C	Within ±5%	1.6 kV
*C* _s2_	CBB	±5%	−40 °C~+100 °C	Within ±5%	1.6 kV

**Table 5 sensors-26-00067-t005:** Comparison of prototype characteristics.

Method	Reference	Extra Components	MOS/Inverter	CC/CV	Variable Output	Efficiency
FM	Proposed	None	1	All	Yes	91.3%
	[29]	n transformers + 1 capacitor	4	CV	Yes	92.1%
Integrated Buck	[28]	4 MOSFETs + 2 inductors	4	No	Yes	89%
Modular Parallel Connection	[20]	None	1	CV	No	92.3%
Clamped Adaptive	[21]	4 diodes + 1 capacitor + 1 inductor	1	All	No	91.2%

**Table 6 sensors-26-00067-t006:** Comparison of prototype output and safety performance.

Method	Reference	Voltage Regulation Range	Voltage Overshoot	System Safety
FM	Proposed	Wide range	No	High
	[28]	Wide range	Yes	Low
Integrated Buck	[29]	A Few Designated	1 Yes, 2 No ^1^	Medium ^2^
Modular Parallel Connection	[20]	A Few Designated	Yes	Low
Clamped Adaptive	[21]	Only Single-Voltage Output	No	High

^1^ represents a three-output prototype, where one output exhibits voltage overshoot while the other two do not. ^2^ indicates that the system safety lies between strict CV output and overshoot-prone output.

## Data Availability

The original contributions presented in this study are included in the article. Further inquiries can be directed to the corresponding author.

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
