# Peer review of "Single-Switch Inverter Modular Parallel Multi-Voltage Levels Wireless Charging System for Robots"

_sensors, 2025, doi:10.3390/s26010067_

Round 1

Reviewer 1 Report

Comments and Suggestions for Authors
  1. The name of the software used to simulate the model should be included in the manuscript.
  2. The contributions of this paper need to be stated more clearly. The limitations of existing works should be highlighted to provide proper context and introduce the novel contributions effectively.
  3. The comparison table needs to be more precise. The names of the methods presented in [28], [29], [20], and [21] should be included, along with additional details such as system size and cost.
  4. How is the efficiency of the proposed inverter calculated?
  5. The authors claim that “the proposed method still achieves a significant reduction in the number of devices compared with approaches reported in other studies”; however, this claim is not justified in the results section or the comparison table.
  6. The statement, ‘The concept of frequency modulation (FM) and parallel unit control further expands the voltage regulation range, prevents excessive voltage rise during charging, and improves system safety,’ is also not supported with sufficient justification in the manuscript.
  7. The statement, ‘Compared to FB or HB multi-parallel inverter circuits, the single-switch multi-parallel inverter uses identical control signals for all switches, thereby minimizing signal-delay effects,’ requires an appropriate reference for support.

Reviewer 2 Report

Comments and Suggestions for Authors • The transition from complex models to the minimal model omits intermediate steps. It does not formally verify the equivalence conditions, especially given that the number of modules affects the overall impedance. This limits the reproducibility of the analysis. • Although inductances, capacitances, and resistances are listed, tolerances, thermal drifts, and experimental conditions (temperature, nominal current) are omitted, even though these factors can shift the resonance and affect the validity of the derived expressions. • It is also not specified how the magnetic parameters (M, Lp, Ls) were measured: the measurement frequency, the current level, or the coil configuration. This makes it difficult to assess the consistency between the model and the actual prototype. • The CC-to-CV detection logic depends on voltage sampling (Gbat) and a 95% threshold (Gs = 0.95Ge). However, it does not detail the sampling frequency, filtering, ADC accuracy, or total digital-loop delay. • The 80–90 kHz range is adopted without explaining why it applies to robots rather than only to electric vehicles. The selection of sub-bands within the range (fxmin, fxmax) is also not justified on the basis of stability, efficiency, or EMI criteria. • The system is described as “load-independent” in both CC and CV, but the validation of this independence is not presented beyond the theoretical model. • The validation of the CC or CV operation is performed by abruptly changing Rload in open-loop mode. This does not reproduce real battery behavior, whose dynamic characteristics are not equivalent to a resistive step. • No temporal response is shown (overshoot, settling time). Only “almost constant” values are reported. Quantitativecharacterization is required. • There is no measurement of switching losses or device temperature at nominal power. • The efficiency curve shows a peak of 91.3%. However, losses are not broken down, measurement uncertainty is not reported, and it is not indicated whether the measurements were performed with a real battery load or an electronic load. • No direct experimental comparison is performed under homogeneous conditions, which weakens the technical relevance of the comparison.

Round 2

Reviewer 1 Report

Comments and Suggestions for Authors

The authors address all the comments.